# Structural basis of anti-SARS-CoV-2 activity of HCQ: specific binding to N protein to disrupt its interaction with nucleic acids and LLPS

## Research Article

**Key words:**
Hydroxychloroquine; liquid–liquid phase separation; NMR spectroscopy; nucleocapsid protein; SARS-CoV-2

**Author for correspondence:**
*Jianxing Song,
E-mail: dbssjx@nus.edu.sg

Mei Dang ⬤ and Jianxing Song* ⬤

Department of Biological Sciences, Faculty of Science, National University of Singapore, 10 Kent Ridge Crescent, Singapore 119260, Singapore

## Abstract

SARS-CoV-2 nucleocapsid (N) protein plays the essential roles in key steps of the viral life cycle, thus representing a top drug target. Functionality of N protein including liquid–liquid phase separation (LLPS) depends on its interaction with nucleic acids. Only the variants with N proteins functional in binding nucleic acids might survive and spread in evolution and indeed, the residues critical for binding nucleic acids are highly conserved. Hydroxychloroquine (HCQ) was shown to prevent the transmission in a large-scale clinical study in Singapore but so far, no specific SARS-CoV-2 protein was experimentally identified to be targeted by HCQ. Here by NMR, we unambiguously decode that HCQ specifically binds NTD and CTD of N protein with Kd of 112.1 and 57.1 μM, respectively to inhibit their interaction with nucleic acid, as well as to disrupt LLPS. Most importantly, HCQ-binding residues are identical in SARS-CoV-2 variants and therefore HCQ is likely effective to different variants. The results not only provide a structural basis for the anti-SARS-CoV-2 activity of HCQ, but also renders HCQ to be the first known drug capable of targeting LLPS. Furthermore, the unique structure of the HCQ-CTD complex suggests a promising strategy for design of better anti-SARS-CoV-2 drugs from HCQ.

**CAMBRIDGE**
UNIVERSITY PRESS

## Introduction

Severe Acute Respiratory Syndrome Coronavirus 2 (SARS-CoV-2) caused the ongoing catastrophic pandemic (Wu *et al.,* 2020), which already led to >219 millions of infections and >4.55 millions of deaths. It belongs to a large family of positive-stranded RNA coronaviruses with ~30 kb genomic RNA (gRNA) packaged with nucleocapsid (N) protein into a membrane-enveloped virion. SARS-CoV-2 has four structural proteins: namely the spike (S) protein that recognises the host-cell receptors angiotensin converting enzyme-2 (ACE2), membrane-associated envelope (E), membrane (M) proteins and N protein.

SARS-CoV-2 N protein is a 419-residue multifunctional protein (Supplementary Fig. S1A), which is composed of the folded N-terminal domain (NTD) over residues 44–173 and C-terminal domain (CTD) over 248–365 (Supplementary Fig. S1B), as well as three intrinsically disordered regions (IDRs) respectively over 1–43, 174–247 and 366–419. Previous studies established that its NTD is an RNA-binding domain (RBD) functioning to bind various viral and host-cell nucleic acids of a broad specificity, which include single- and double-stranded RNA/DNA of diverse sequences, while CTD acts to dimerize/oligomerize to form high-order structures (Cong *et al.,* 2020; Dinesh *et al.,* 2020; Peng *et al.,* 2020; Ross, 2020; Redzic *et al.,* 2021; Zinzula *et al.,* 2021). Coronavirus N proteins appear to have two major categories of functions: while their primary role is to assemble gRNA and N protein to form the viral gRNA-Nprotein (vRNP) complex into the new virions at the final stage of the infection, they also act to suppress the immune system of the host cell and to hijack cellular machineries to achieve the replication of the virus, such as to interfere in the formation of stress granules (SGs) and to localise gRNA onto the replicase-transcriptase complexes (RTCs) (Chen *et al.,* 2007; Carlson *et al.,* 2020; Cong *et al.,* 2020; Dinesh *et al.,* 2020; Iserman *et al.,* 2020; Peng *et al.,* 2020; Perdikari *et al.,* 2020; Ross, 2020; Savastano *et al.,* 2020; Dang *et al.,* 2021; Lu *et al.,* 2021; Redzic *et al.,* 2021; Zinzula *et al.,* 2021).

Very recently, liquid–liquid phase separation (LLPS), the emerging principle for commonly organising the membrane-less organelles (MLOs) or compartments critical for cellular physiology and pathology (Hyman *et al.,* 2014; Patel *et al.,* 2017; Shin and Brangwynne, 2017), has been identified as the key mechanism underlying the diverse functions of SARS-CoV-2 N protein (Carlson *et al.,* 2020; Iserman *et al.,* 2020; Perdikari *et al.,* 2020; Savastano *et al.,* 2020; Dang *et al.,* 2021; Lu *et al.,* 2021). Nevertheless, almost all identified functions of N proteins so far appear to be dependent on its capacity in binding various viral and host-cell RNA/DNA of both specific and non-specific sequences. For example, LLPS of N protein is mainly driven by its dynamic and multivalent interactions with nucleic acids similar to what was previously observed on human FUS and TDP-43 proteins (Kang *et al.,* 2019a; Song, 2021), while N protein with nucleic acids completely removed lacks the intrinsic capacity in LLPS (Carlson *et al.,* 2020; Iserman *et al.,* 2020;

Savastano *et al.*, 2020; Dang *et al.*, 2021). In particular, although the detailed mechanism still remains poorly understood, the final package of the RNA genome into new virions certainly requires the complex but precise interaction between gRNA and N protein, which should be extremely challenging for SARS-CoV-2 with such a large RNA genome (~30 kb). In this context, any small molecules capable of intervening in the interaction of N protein with nucleic acids are anticipated to significantly modulate most, if not all, key steps of the viral life cycle, some of which may thus manifest the anti-SARS-CoV-2 activity.

Hydroxychloroquine (HCQ), an antimalarial drug (Fig. 1*c*), has been extensively proposed for clinically treating the SARS-CoV-2 pandemic (Roldan *et al.*, 2020; Satarker *et al.*, 2020; Seet *et al.*, 2021). Particularly, a large-scale clinical study in Singapore showed that oral HCQ can indeed prevent the spread of SARS-CoV-2 in the high transmission environments (Seet *et al.*, 2021). However, the exact mechanisms for the anti-SARS-CoV-2 activity of HCQ remain highly elusive and so far, all the proposed action sites for HCQ are on the host cells, which include the interference in the endocytic pathway, blockade of sialic acid receptors, restriction of pH mediated S protein cleavage at the ACE2 binding site and prevention of cytokine storm (Roldan *et al.*, 2020; Satarker *et al.*, 2020). In particular, to date no SARS-CoV-2 protein has been experimentally identified to be targeted by HCQ.

Very unexpectedly, here our residue-specific characterisation by NMR spectroscopy decrypted that: (a) HCQ specifically binds both NTD and CTD with Kd of 112.1 and 57.1 μM, respectively to inhibit their interactions with nucleic acids. (b) HCQ has no capacity in inducing LLPS of N protein but is able to dissolve LLPS of N protein induced by nucleic acids, which is underlying the key steps of the viral life cycle. Therefore, the results not only provide a structural basis for the anti-SARS-CoV-2 activity of HCQ, but to the best of our knowledge, render HCQ to be the first known drug capable of targeting LLPS. Furthermore, the unique structure of the HCQ-CTD complex offers a promising strategy for further design of better anti-SARS-CoV-2 drugs starting from HCQ.

## Results

### HCQ specifically bind NTD to inhibit its interaction with nucleic acid

Sequences of N proteins particularly over NTD and CTD have very high identity between SARS-CoV-1 and SARS-CoV-2 (Supplementary Fig. S6). Previously the interaction between NTD of SARS-CoV-1 N protein and nucleic acids including single- and double-stranded RNA/DNA has been extensively studied by a variety of biophysical methods, but no high-resolution structure could be determined for the NTD-nucleic-acid complex (Chen *et al.*, 2007; Chang *et al.*, 2014). Very recently, NTD of SARS-CoV-2 N protein in the free state has been determined by NMR and the structures of its complex with RNA fragments were constructed by HADDOCK (Dinesh *et al.*, 2020). Further NMR studies revealed that upon binding to nucleic acids, μs–ms dynamics have been provoked over a large portion of NTD residues, thus rationalising the failure in determining the complex structure (Redzic *et al.*, 2021).

NTD has been shown to utilise a large positively charged surface to bind various RNA/DNA of diverse sequences and structures with dissociation constants of ~μM. Depending on the types, lengths, sequences and structures of nucleic acids, the binding of nucleic acids can trigger very different changes of NMR peaks of NTD which include the shift or/and disappearance of NMR peaks. For example, short nucleic acid fragments with the relatively low affinity mainly induces the shift of NMR peaks while long fragments with the high affinity triggers the broadening/disappearance of NMR peaks. In fact, this was also extensively observed on other nucleic-acid-binding domains such as RNA-recognition motif (RRM) domain of FUS (Kang *et al.*, 2019*b*).

In the present study, we employed NMR spectroscopy to assess the interaction between NTD/CTD and nucleic acid, because NMR is a very powerful tool for the residue-specific characterisation of the weak binding associated with unstable and aggregation-prone protein samples, which are thus not amenable to investigations by other biophysical methods such as ITC capable of inducing unfolding by shear flow (Qin *et al.*, 2008; Williamson, 2013; Qin *et al.*, 2015; Kang *et al.*, 2019*b*). With regard to nucleic acid, although previously we used the non-specific 24-mer poly(dA) which has a relatively low binding affinity and thus mainly induced the shift of NMR HSQC peaks of SARS-CoV-2 N protein domains (Dang *et al.*, 2021), here in order to better mimic the biologically relevant environment, we used S2m, a 32-mer stem-loop II motif (S2m) derived from the SARS-CoV-2 viral gRNA (Supplementary Fig. S1C), which is a highly conserved sequence among all coronaviruses (Aldhumani *et al.*, 2021). Most importantly, previously S2m has been extensively utilised to identify nucleic-acid-binding domains of SARS-CoV-1 beyond N protein (Chen *et al.*, 2007). *In vitro* SARS-CoV-1 S2m was shown to fold into a flexible structure with some regions single-stranded and other double-stranded (Aldhumani *et al.*, 2021), while its RNA and DNA forms both could bind N protein (Chen *et al.*, 2007). In the present study, we used the DNA form of S2m which is much more chemically stable than the RNA form because we previously found that RNAs could get degraded during NMR titrations due to the easy contamination of ribonuclease (Qin *et al.*, 2014; Kang *et al.*, 2019*a*).

Indeed, as monitored by NMR, S2m could bind NTD characteristic of extensive broadening of HSQC peaks (Supplementary Fig. S2). At 1:2.5 (NTD:S2m), a large portion of HSQC peaks became disappeared and further addition of S2m led to no significant change of HSQC spectra. This observation implies that the binding of NTD with S2m triggered significant conformational dynamics over the intermediate NMR time scale (Qin *et al.*, 2008; Williamson, 2013; Qin *et al.*, 2015; Kang *et al.*, 2019*b*), completely consistent with the previous reports on SARS-CoV-2 (Dinesh *et al.*, 2020; Redzic *et al.*, 2021) and SARS-CoV-1 (Chen *et al.*, 2007; Chang *et al.*, 2014) NTD.

Subsequently, we evaluated the binding of HCQ to NTD by stepwise addition of HCQ as monitored by NMR. Strikingly, only a small set of HSQC peaks were shifted and the shifting process was largely saturated at 1:15 (NTD:HCQ) (Fig. 1*a*), unambiguously revealing that HCQ does specifically bind NTD. All NMR HSQC spectra in the presence of HCQ at different ratios were assigned and only eleven residues including Trp52, Leu56, His59, Ser79, Gly147, Ile157, Val158, Leu159, Thr166, Ala173 and Glu174 were significantly perturbed upon adding HCQ (Fig. 1*b*). Although these residues are distributed over the whole NTD sequence, they become clustered together in the NTD structure (Fig. 2*a*). We then conducted fitting of the NMR shift tracings by the well-established procedure (Williamson, 2013; Kang *et al.*, 2019*b*; Dinesh *et al.*, 2020; Perdikari *et al.*, 2020) of 11 residues to obtain their residue-specific Kd values (Supplementary Table S1 and Fig. 2*b*) with the average Kd of 112.1 ± 32.2 μM. Furthermore, with NMR-derived constraints, the structure of the HCQ-NTD complex was constructed (Fig. 2*c*) by the well-established HADDOCK program

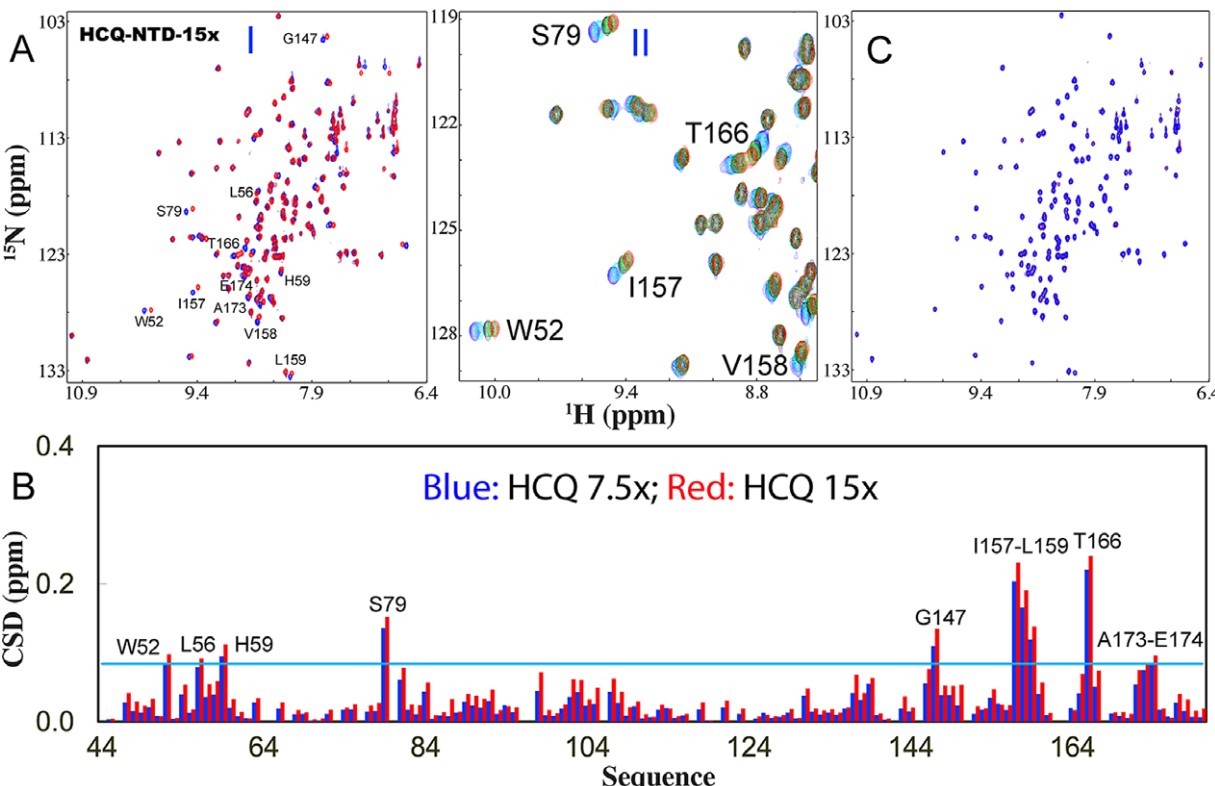

**Fig. 1.** HCQ specifically binds NTD to inhibit its interaction with nucleic acid. (*a*) (I) Superimposition of HSQC spectra of NTD in the free state (blue) and in the presence of HCQ at 1:15 (NTD:HCQ) (red). The assignments of the significantly perturbed residues are labelled. (II) Zoom of HSQC spectra of NTD in the absence (blue) and in presence of HCQ at 1:1.88 (cyan); 1:3.75 (black); 1:7.5 (green); 1:15 (red). (*b*) Residue-specific chemical shift difference (CSD) of NTD in the presence of HCQ at 1:7.5 (blue) and 1:15 (red) (NTD:HCQ). The significantly perturbed residues are labelled, which are defined as those with the CSD values at 1:15 > 0.084 (average value + one standard deviation) (cyan line). (*c*) Superimposition of HSQC spectra of NTD only in the presence of HCQ at 1:15 (red) and with additional addition of S2m at 1;2.5 (NTD:S2m) (blue).

(Dominguez *et al.,* 2003; Qin *et al.,* 2008; Kang *et al.,* 2019b). In the complex structure, the binding pocket of HCQ is located within the conserved negatively charged surface for NTD to bind various nucleic acids (Chang *et al.,* 2014; Dinesh *et al.,* 2020; Redzic *et al.,* 2021).

We then asked a question whether the HCQ binding can interfere in the binding of NTD with S2m. To address this question, we added S2m to the NTD sample with the pre-existence of HCQ at 1:15 and found no significant change of HSQC spectra even up to 1:2.5 (NTD:S2m) (Fig. 1c). This result suggests that the HCQ-bound NTD was inhibited for further binding to S2m. To confirm this, we also prepared the NTD sample pre-saturated with S2m at 1:2.5 and then stepwise added HCQ into this sample (Supplementary Fig. S3). Interestingly, even upon adding HCQ at 1:3.75 (NTD:HCQ), many disappeared HSQC peaks due to being bound with S2m became restored and at 1:15, all HSQC peaks could become detected which are very similar to those of the NTD sample only in the presence of HCQ at 1:15 (NTD:HCQ). This result unambiguously indicates that HCQ is indeed able to displace S2m from being bound with NTD.

### HCQ specifically bind CTD to inhibit its interaction with nucleic acid

In parallel, we titrated the $^{15}$N-labelled CTD with S2m and S2m was also able to bind CTD characterised by significant broadening of HSQC peaks (Supplementary Fig. S4). Most HSQC peaks became disappeared even at 1:1 (CTD:S2m) and further addition of S2m

induced no significant change. This is similar to what was previously reported on SARS-CoV-1 N protein by the gel-shift assay: CTD binds S2m with the affinity higher than that of NTD (Chen *et al.,* 2007).

Subsequently, we stepwise added HCQ into the CTD sample and again only a small set of HSQC peaks became shifted. At 1:7.5 (CTD:S2m) the shifting process was largely saturated (Fig. 3a). Detailed analysis of the spectra at different HCQ concentrations revealed that only seven CTD residues including Gln281, Thr282, Thr325, Thr329, Trp330, Ala336 and Ile337 were significantly perturbed upon adding HCQ (Fig. 3b), which are located over the whole sequence but clustered together to form two pockets in the dimeric CTD structure (Fig. 4a). The residue-specific Kd values of seven residues were successfully obtained (Supplementary Table S1 and Fig. 4b) with the average Kd of $52.1 \pm 17.6\ \mu M$. The structure of the HCQ-CTD complex was also constructed from NMR-derived constraints in which the dimeric CTD is bound with two HCQ molecules (Fig. 4c). Interestingly, the binding of HCQ appears to be mainly driven by the insertion of the aromatic rings of HCQ into the pockets of the cleft on the same side of the dimeric CTD structure.

We also assessed the interplay of HCQ and S2m in binding CTD. Briefly, we added S2m to the CTD sample in the pre-existence of HCQ at 1:7.5 but found no significant change of HSQC spectra even up to 1:1 (CTD:S2m) (Fig. 3c). This observation suggests that the HCQ-bound CTD is also blocked for binding with S2m. We also prepared the CTD sample pre-saturated with S2m at 1:1 into which HCQ was stepwise added (Supplementary Fig. S5). Interestingly,

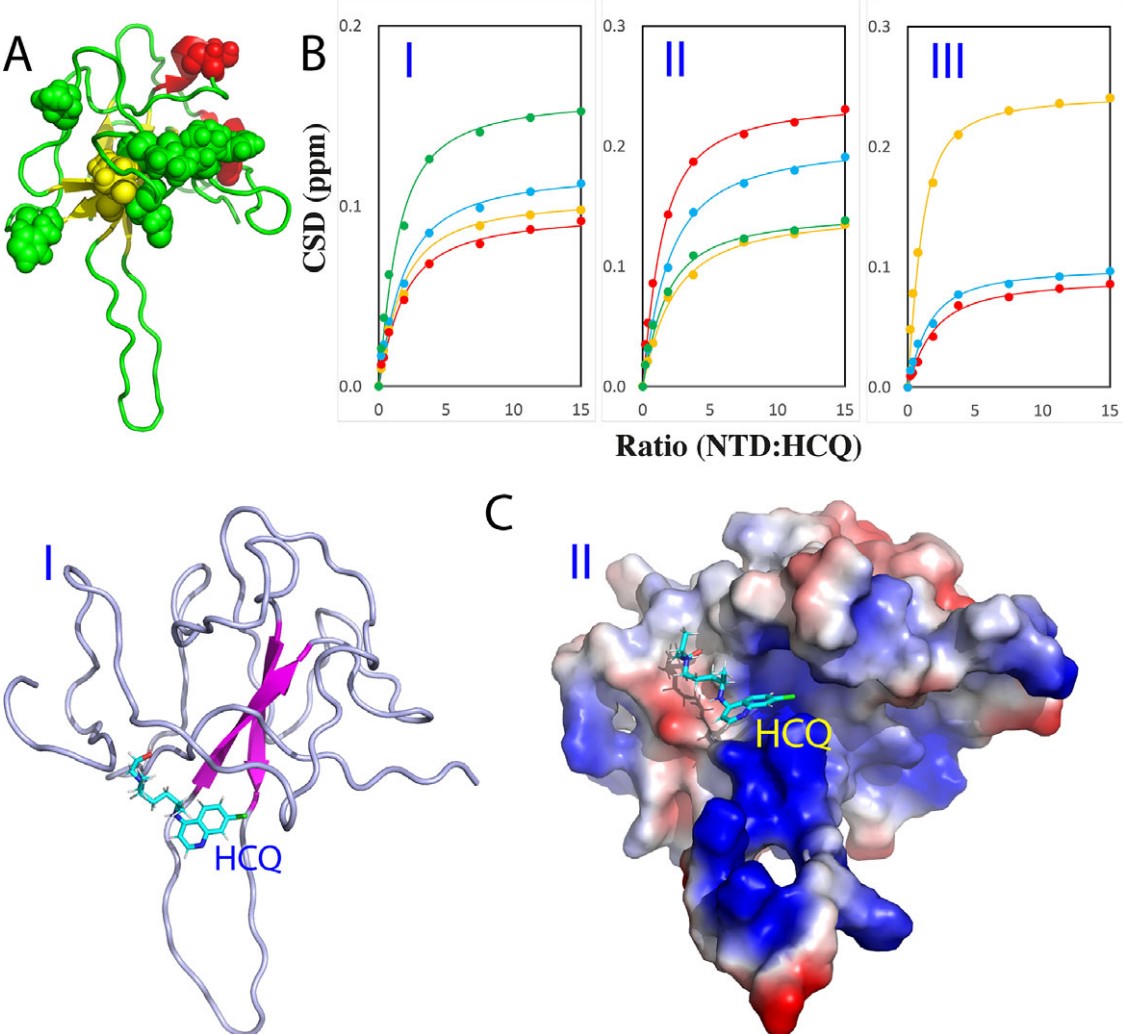

**Fig. 2.** NMR characterisation of the binding of HCQ to NTD. (*a*) Eleven significantly perturbed residues of NTD upon binding to HCQ. (*b*) Fitting of 11 NTD residue-specific dissociation constant (Kd): experimental (dots) and fitted (lines) values for the CSDs induced by addition of HCQ at different ratios. (I) Trp52 (brown), Leu56 (red), His59 (cyan) and Ser79 (green). (II) Gly147 (brown), Ile157 (red), Val158 (cyan) and Leu159 (green). (III) Thr166 (brown), Ala173 (red) and Glu174 (cyan). (*c*) Structure of the HCQ-NTD complex with HCQ in sticks and NTD in ribbon (I) and in electrostatic potential surface (II).

upon adding HCQ at 1:1.8 (NTD:HCQ), many disappeared HSQC peaks became restored and at 1:7.5, all HSQC peaks could become detected which are highly similar to those of the CTD sample only in the presence of HCQ at 1:7.5 (CTD:HCQ).

NMR results here suggest that: (a) similar to what was observed on NTD, S2m also binds CTD and induces conformational dynamics over the intermediate NMR time scale, thus leading to significant disappearance of NMR peaks, which rationalises the failure of the previous effort to determine the crystal structure of the S2m-CTD complex of SARS-CoV-1 N protein (Chen *et al.,* 2007); (b) despite the lack of the S2m-CTD structure to provide the mechanism at an atomic resolution, here the results of NMR competition experiments unambiguously reveals that HCQ can displace S2m from being bound with CTD.

### HCQ dissolves LLPS of N protein induced by nucleic acid

Very recently, the SARS-CoV-2 N protein was shown to function through LLPS, which was induced by dynamic and multivalent interactions with various nucleic acids including viral and host-cell RNA and DNA (Carlson *et al.,* 2020; Iserman *et al.,* 2020; Perdikari *et al.,* 2020; Savastano *et al.,* 2020; Dang *et al.,* 2021; Lu *et al.,* 2021). Here we thus asked a question whether HCQ has any effect on LLPS of SARS-CoV-2 N protein. To address this, we first titrated S2m into the isolated NTD and CTD samples as monitored by measuring turbidity (absorption at 600 nm) and imaging by differential interference contrast (DIC) microscopy as we previously conducted on SARS-CoV-2 N protein (Dang *et al.,* 2021) and FUS (Kang *et al.,* 2019*a*). However, no liquid droplets or aggregate was observed on the isolated NTD and CTD samples upon adding S2m.

By contrast, consistent with the previous reports including by us with specific and non-specific nucleic acid fragments of different lengths (Carlson *et al.,* 2020; Iserman *et al.,* 2020; Perdikari *et al.,* 2020; Savastano *et al.,* 2020; Dang *et al.,* 2021; Lu *et al.,* 2021), S2m also imposed the biphasic effect on LLPS of the full-length N protein: induction at low ratios and dissolution at high ratios. Briefly, while the N protein showed no LLPS in the free state, LLPS was induced upon addition of S2m as evidenced by turbidity and DIC imaging. At 1:0.75 (Nprotein:S2m), the turbidity reached the highest value of 1.92 (I of Fig. 5*a*) and many liquid droplets were

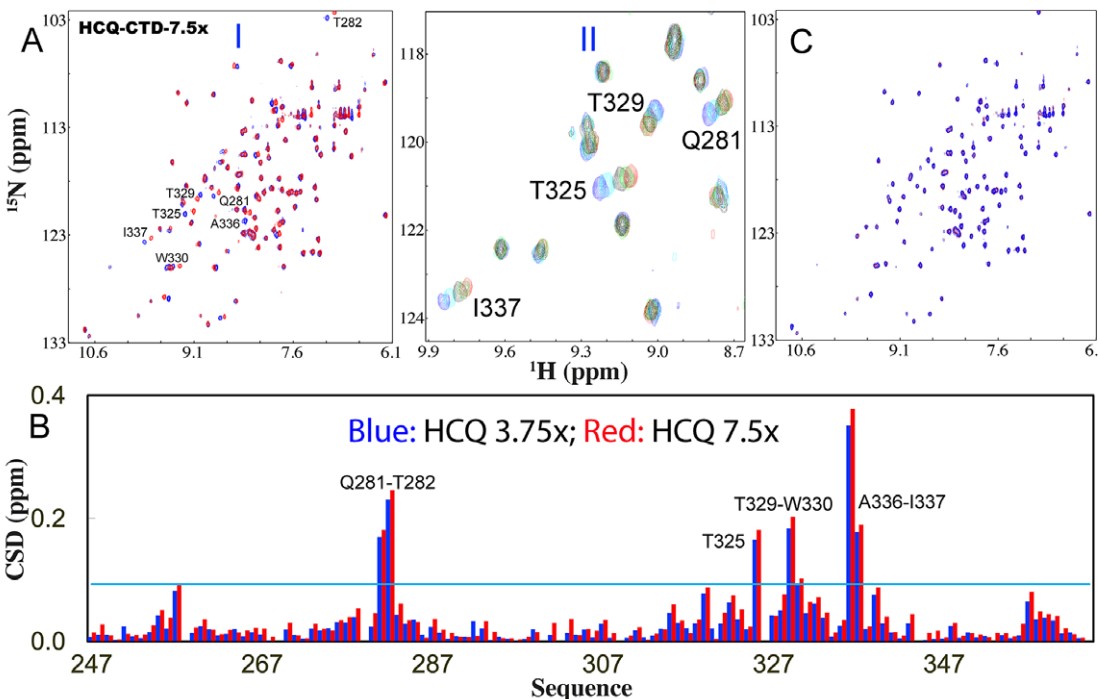

**Fig. 3.** HCQ specifically binds CTD to inhibit its interaction with nucleic acid. (*a*) (I) Superimposition of HSQC spectra of CTD in the free state (blue) and in the presence of HCQ at 1:7.5 (CTD:HCQ) (red). (II) Zoom of HSQC spectra of CTD in the absence (blue) and in presence of HCQ at 1:0.925 (cyan); 1:1.86 (black); 1:3.75 (green); 1:7.5 (red). The assignments of the significantly perturbed residues are labelled. (*b*) Residue-specific chemical shift difference (CSD) of CTD in the presence of HCQ at 1:3.75 (blue) and 1:7.5 (red) (CTD:HCQ). The significantly perturbed residues are labelled, which are defined as those with the CSD values at 1:7.5 > 0.093 (average value + one standard deviation) (cyan line). (*c*) Superimposition of HSQC spectra of CTD only in the presence of HCQ at 1:7.5 (red) and with additional addition of S2m at 1:1 (CTD:S2m) (blue).

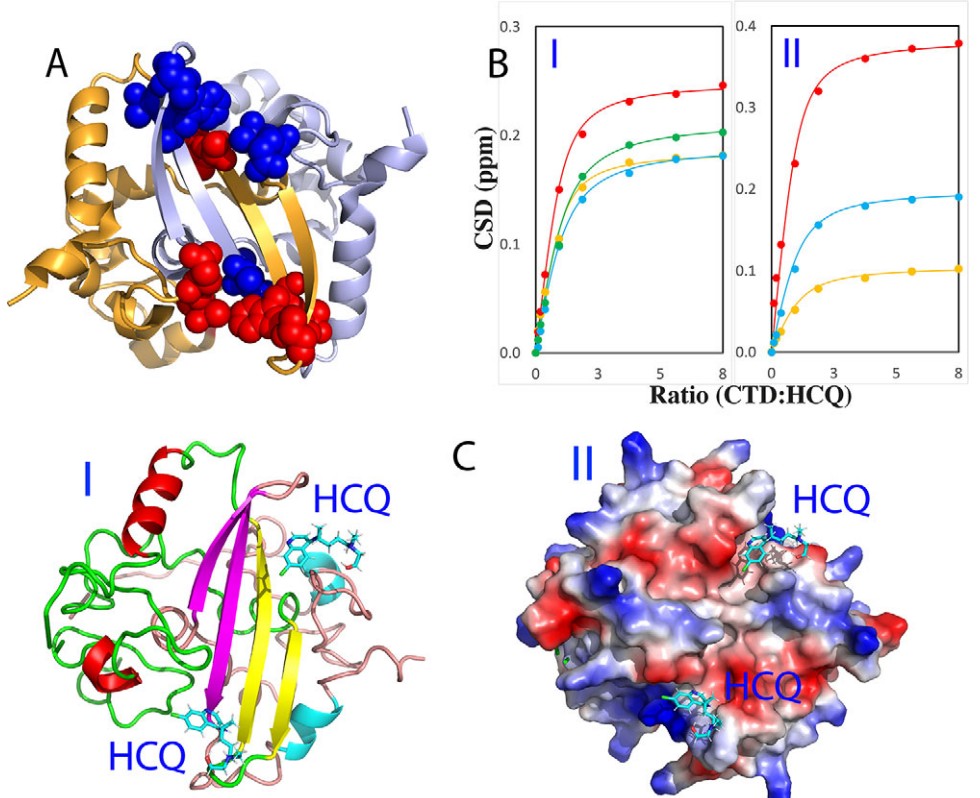

**Fig. 4.** NMR characterisation of the binding of HCQ to CTD. (*a*) Seven significantly perturbed residues of CTD upon binding to HCQ. (*b*) Fitting of seven CTD residue-specific dissociation constant (Kd): experimental (dots) and fitted (lines) values for the CSDs induced by addition of HCQ at different ratios. (I) Gln281 (brown), Thr282 (red), Thr325 (cyan) and Thr329 (green). (II) Trp330 (brown), Ala336 (red) and Ile337 (cyan). (*c*) Structure of the HCQ-CTD complex with HCQ in sticks and CTD in ribbon (I) and in electrostatic potential surface (II).

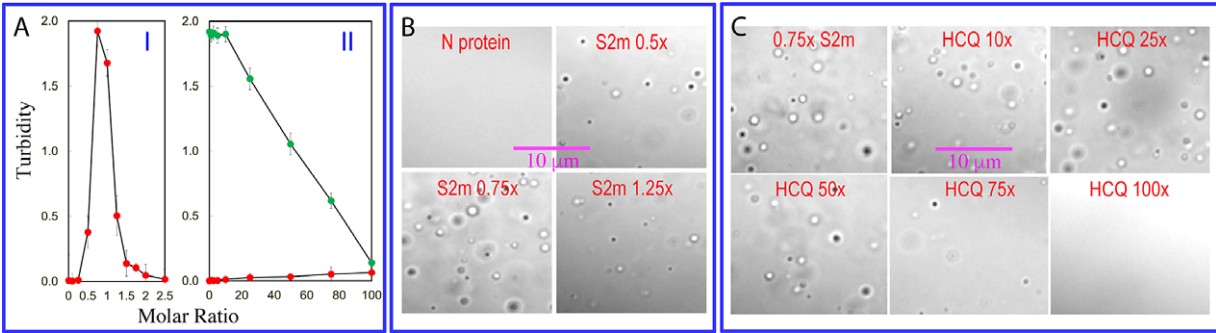

**Fig. 5.** HCQ disrupts LLPS of SARS-CoV-2 N protein. (a) Turbidity curves of N protein in the presence of S2m at different ratios (I). (II) Turbidity curves of N protein in the absence (red) or in the presence of S2m at 1:0.75 (green) with additional addition of HCQ at different ratios. (b) DIC images of N protein in the presence of S2m at different ratios. (c) DIC images of N protein in the presence of S2m at 1:0.75 with additional addition of HCQ at different ratios.

formed (Fig. 5b). Further addition of S2m dissolved LLPS and at 1:1.5 all liquid droplets were dissolved.

We then stepwise added HCQ into the N protein sample under the same conditions but no LLPS occurred even with the ratio up to 1:100 (II of Fig. 5a), indicating that HCQ is incapable of inducing LLPS. We then prepared a phase separated N protein sample with the pre-presence of S2m at 1:0.75 and subsequently added HCQ into this sample in a stepwise manner, as monitored by turbidity (II of Fig. 5a) and DIC imaging (Fig. 5c). At the ratios <1:20, HCQ has no significant effect on LLPS. However, at ratios >1:20, HCQ monotonically dissolved liquid droplets as evidenced by the gradual reduction of turbidity and disappearance of liquid droplets. Strikingly, at 1:100 (Nprotein:HCQ), liquid droplets were completely dissolved. Previously it was established that the dynamic and multivalent binding of nucleic acids to SARS-CoV-2 N protein plays a key role in driving its LLPS (Chen *et al.,* 2007; Peng *et al.,* 2020; Perdikari *et al.,* 2020; Ross, 2020; Lu *et al.,* 2021; Redzic *et al.,* 2021). In this context, the present results together decode that HCQ acts to dissolve LLPS of SARS-CoV-2 N protein most likely by displacing the nucleic acid from being bound with both NTD and CTD.

## Discussion

So far, great efforts have been dedicated to successfully developing the spike-based vaccines to combat the pandemic. Nevertheless, many challenges still remain to completely terminate the pandemic, which include the rapidly emerging antibody-resistance variants (Wang *et al.,* 2021), the adverse effects of the spike protein (Lei *et al.,* 2021) and even its antibody (Hoepel *et al.,* 2021). Most seriously, SARS-CoV-2 spike protein has been identified to provoke antibody-dependent enhancement (ADE) of infection (Lee *et al.,* 2021; Liu *et al.,* 2021) while its RNA fragments might get integrated into human genome (Zhang *et al.,* 2021). Therefore, any small molecule drugs that directly target SARS-CoV-2 proteins to disrupt its life cycle are extremely valuable and urgently demanded to finally terminate the pandemic.

Out of SARS-CoV-2 proteins, N protein is the only one which plays the essential roles in almost all key steps of the viral life cycle, thus representing a top drug target. Mechanistically, almost all functions of N protein including LLPS appear to depend on its capacity in interacting with a variety of viral and host-cell nucleic acids of diverse types, sequences and structures. Therefore, in evolution only the SARS-CoV-2 variants with their N protein functional in binding nucleic acids can survive and spread. Indeed,

as shown in Supplementary Fig. S6, the sequences of NTD and CTD critical for binding nucleic acids are not only highly conserved in the different variants of SARS-CoV-2, but also in SARS-CoV-1. In this context, any small molecules capable of blocking the interaction of N protein with nucleic acids would disrupt the viral life cycle of these variants.

Here, for the first time, HCQ, a safe drug recommended by WHO to treat other diseases (Roldan *et al.,* 2020; Satarker *et al.,* 2020; Seet *et al.,* 2021), has been decrypted to specifically bind NTD and CTD of SARS-CoV-2 N protein to inhibit its interactions with nucleic acids as well as to dissolve LLPS. This finding not only provided an acting mechanism for the anti-SARS-CoV-2 activity of HCQ, but also validated that SARS-CoV-2 N protein and its LLPS are indeed druggable by small molecules, thus opening up a new direction for further development of anti-SARS-CoV-2 drugs by targeting N protein and its LLPS. Previously it was shown that HCQ could inhibit the maturation of SARS-CoV-2 virions, but this was proposed to result from the HCQ-induced changes of host-cell structures/conditions such as pH (Roldan *et al.,* 2020; Satarker *et al.,* 2020). The current results suggest that the ability of HCQ to specifically bind NTD and CTD to disrupt LLPS of N protein may at least partly contribute to the inhibition of the maturation of SARS-CoV-2 virions. To the best of our knowledge, HCQ is the first drug which has been revealed to target LLPS, thus bearing the unprecedented implications for further design of drugs in general by modulating LLPS, whose roles in underlying various human diseases are starting to be recognised (Hyman *et al.,* 2014; Patel *et al.,* 2017; Shin and Brangwynne, 2017; Kang *et al.,* 2019a; Carlson *et al.,* 2020; Iserman *et al.,* 2020; Perdikari *et al.,* 2020; Savastano *et al.,* 2020; Dang *et al.,* 2021; Lu *et al.,* 2021; Song, 2021).

At the fundamental level of molecular interactions, the core mechanisms for LLPS induced by nucleic acids appear to be relatively conserved (Hyman *et al.,* 2014; Patel *et al.,* 2017; Shin and Brangwynne, 2017; Kang *et al.,* 2019a; Carlson *et al.,* 2020; Iserman *et al.,* 2020; Perdikari *et al.,* 2020; Savastano *et al.,* 2020; Dang *et al.,* 2021; Lu *et al.,* 2021; Song, 2021). In particular, the significantly perturbed residues of NTD and CTD upon binding HCQ are completely identical in all variants of SARS-CoV-2, as well as even highly conserved in SARS-CoV-1 (Supplementary Fig. S6). Consequently, HCQ which acts to block the nucleic-acid-binding and LLPS of N proteins is anticipated to be likely effective to most, if not all, variants of SARS-CoV-2. On the other hand, emerging evidence suggests that the nucleic-acid-induced LLPS is not only essential for the life cycles of coronaviruses, but might be generally critical for other virus-host interactions (Brocca *et al.,* 2020; Guseva *et al.,*

2020; Monette *et al.,* 2020; Salladini *et al.,* 2021; Scoca *et al.,* 2021). This might thus explain the puzzling observations that HCQ was not only effective in treatment of SARS-CoV-2 and SARS-CoV-1, but also to Dengue/Zika infections (Roldan *et al.,* 2020; Satarker *et al.,* 2020). In the future it would be also of significant interest to explore whether the mechanisms of HCQ in treating other diseases are also related to its capacity in intervening in LLPS.

The relatively low binding affinities of HCQ to NTD and CTD of SARS-CoV-2 N protein might rationalise the reports that HCQ is effective in preventing the infection as well as in treating the infection at the early stage (Roldan *et al.,* 2020; Satarker *et al.,* 2020; Seet *et al.,* 2021), because at the initial stage of infection, the number of viral gRNA and N protein should be very low in the host cell based on the estimation that in one SARS-CoV-2 virion, ~730–2200 copies of N protein form the complex with one 30-kb gRNA (Bar-On *et al.,* 2020).

The unique HCQ-CTD structure (Fig. 4) in which two HCQ molecules use the aromatic rings to insert into two binding pockets of a cleft on the same side of the dimeric CTD might offer a promising strategy for further design of anti-SARS-CoV-2 drugs with better affinity and specificity. Briefly, by introducing the proper groups to link two HCQ aromatic rings, the bivalent or even multivalent binders might be engineered, whose Kd value is the time of the Kd values of individual groups (Song and Ni, 1998). In particular, those bivalent/multivalent binders are also anticipated to have the high specificity because the probability for human proteins to have such unique two HCQ binding pockets on the dimeric CTD should be extremely low. In fact, we have already constructed such a molecule in silico designated as DiHCQ by linking two HCQ molecules (Supplementary Fig. S7A) and conducted the docking to the dimeric CTD. Indeed, two aromatic rings of DiHCQ do bind the pockets in the manner similar to those of two individual HCQ molecules (Supplementary Fig. S7B).

Due to the extreme urgency to combat the pandemic, here we propose to integrate the combinatorial chemistry as well as biophysical methods including experimental such as NMR and computational such as MD simulations (Shi *et al.,* 2011; Lim *et al.,* 2020) to design bivalent or even multivalent small molecules starting from HCQ to obtain more efficient anti-SARS-CoV-2 drugs, which bind the dimeric CTD of N protein not only by its two aromatic rings, but also by the linker groups.

## Materials and methods

### Preparation of recombinant SARS-CoV-2 nucleocapsid as well as its NTD and CTD

The gene encoding 419-residue SARS-CoV-2 N protein was purchased from a local company (Bio Basic Asia Pacific Pte. Ltd, Singapore), which was cloned into an expression vector pET-28a with a TEV protease cleavage site between N protein and N-terminal 6xHis-SUMO tag used to enhance the solubility. The DNA fragments encoding its NTD (44–180) and CTD (247–364) were subsequently generated by PCR rection and cloned into the same vector (Dang *et al.,* 2021).

The recombinant N protein and its NTD/CTD were expression in *E. coli* cells BL21 with IPTG induction at 18°C. Both proteins were found to be soluble in the supernatant. For NMR studies, the bacteria were grown in M9 medium with addition of $(^{15}NH_4)_2SO_4$ for $^{15}N$-labelling. The recombinant proteins were first purified by $Ni^{2+}$-affinity column (Novagen) under native conditions and subsequently in-gel cleavage by TEV protease was conducted. The

eluted fractions containing the recombinant proteins were further purified by FPLC chromatography system with a Superdex-200 column for the full-length and a Superdex-75 column for NTD and CTD (Dang *et al.,* 2021). The purity of the recombinant proteins was checked by SDS-PAGE gels and NMR assignment for both NTD and CTD. Hydroxychloroquine (HCQ) sulphate was purchased from Merck (HPLC purified, >95%). Protein concentration was determined by spectroscopic method in the presence of 8 M urea (Pace *et al.,* 1995).

### LLPS imaged by DIC microscopy

The formation of liquid droplets was imaged on 50 μl of the N protein samples by DIC microscopy (OLYMPUS IX73 Inverted Microscope System with OLYMPUS DP74 Colour Camera) as previously described (Kang *et al.,* 2019a; Dang *et al.,* 2021). The N protein samples were prepared at 10 μM in 25 mM HEPES buffer (pH 7.0) with 70 mM KCl (buffer 1), while NTD and CTD samples were prepared at 100 and 200 μM, respectively in 10 mM sodium phosphate buffer (pH 7.0) in the presence of 150 mM NaCl (buffer 2) for NMR studies. HCQ at 10 mM was dissolved in buffer 1 for LLPS or buffer 2 for NMR binding studies with the final pH adjusted to 7.0.

### NMR characterizations of the binding of HCQ to NTD and CTD

NMR experiments were conducted at 25°C on an 800 MHz Bruker Avance spectrometer equipped with pulse field gradient units and a shielded cryoprobe as described previously (Sattler *et al.,* 1999; Kang *et al.,* 2019a; Dang *et al.,* 2021). For NMR HSQC titrations with HCQ, two dimensional $^1H$–$^{15}N$ NMR HSQC spectra were collected on the $^{15}N$-labelled NTD at 100 μM or CTD at 200 μM in the absence and in the presence of HCQ at different ratios. NMR data were processed with NMRPipe (Delaglio *et al.,* 1995) and analysed with NMRView (Johnson and Blevins, 1994).

### Calculation of CSD and data fitting

Sequential assignments were achieved based on the deposited NMR chemical shifts for NTD (Dinesh *et al.,* 2020) and CTD (Korn *et al.,* 2021). To calculate chemical shift difference (CSD), HSQC spectra collected without and with HCQ at different concentrations were superimposed. Subsequently, the shifted HSQC peaks were identified and further assigned to the corresponding NTD and CTD residues. The chemical shift difference (CSD) was calculated by an integrated index with the following formula (Williamson, 2013; Kang *et al.,* 2019a, 2019b; Dang *et al.,* 2021):

$$CSD = \left( \left( \Delta^1H \right)^2 + \left( \Delta^{15}N \right)^2/4 \right)^{1/2}.$$

In order to obtain residue-specific dissociation constant (Kd), we fitted the shift traces of the NTD and CTD residues with significant shifts (CSD > average + STD) by using the one binding site model with the following formula as we previously performed (Williamson, 2013; Kang *et al.,* 2019a, 2019b; Dang *et al.,* 2021):

$$CSD_{obs} = CSD_{max}$$
$$\left\{ ([P]+[L]+Kd) - \left[ ([P]+[L]+Kd)^2 - 4[P][L] \right]^{1/2} \right\}/2[P].$$

Here, [P] and [L] are molar concentrations of NTD/CTD and ligands (HCQ), respectively.

## Molecular docking

The structures of the HCQ-NTD and HCQ-CTD complex were constructed by use of the well-established HADDOCK software (Dominguez *et al.*, 2003; Qin *et al.*, 2014; Dang *et al.*, 2021) in combination with crystallography and NMR system (CNS) (Brunger *et al.*, 1998), which makes use of CSD data to derive the docking that allows various degrees of flexibility. Briefly, HAD-DOCK docking was performed in three stages: (Wu *et al.*, 2020) randomisation and rigid body docking; (Dinesh *et al.*, 2020) semi-flexible simulated annealing; and (Zinzula *et al.*, 2021) flexible explicit solvent refinement.

The NMR structure (Dinesh *et al.*, 2020) of NTD (PDB ID of 6YI3) and crystal structure (Zinzula *et al.*, 2021) of CTD (PDB ID of 6YUN) were used for docking to HCQ. The HCQ-NTD and HCQ-CTD structures with the lowest energy score were selected for the detailed analysis and display by Pymol (The PyMOL Molecular Graphics System).

**Acknowledgement.** This study is supported by Ministry of Education of Singapore (MOE) Tier 1 Grants R-154-000-B92-114 to J.S.

**Conflict of interest.**   The authors declare no conflicts of interest.

**Author contributions.** Conceived the research: J.S.; Performed research and analysed data: M.D., J.S.; Acquired funding: J.S.; Wrote manuscript: J.S.

**Supplementary Materials.**   To view supplementary material for this article, please visit http://dx.doi.org/10.1017/qrd.2021.12.

**Open Peer Review.**   To view the open peer review materials for this article, please visit http://doi.org/10.1017/qrd.2021.12.

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
