## [Reviewer Report]

*Comments to Author*: This is a very well conducted study with interesting results. The structures are well presented and the NMR work carefully done.

My minor revision suggestion is about some minor language problems- the ms should be scrutinized again before publication.

examples of some problems:

p.6, 2nd para., line 4:... NTD characteristic of broadening of HSQC... (difficult to understand)

p. 10, 1st para., line 8:...valueless... (do you mean valuable?)

p. 11, 3rd para., line 2:.. effective... (do you really mean effective?)

In figures 2 and 4 you call one color "brown", but in my print it more looks like yellow (?)

---

## [Reviewer Report]

*Comments to Author*: This paper describes experiments, mostly using NMR titrations, that are used to conclude that hydroxychloroquine (HCQ) binds to both the N-terminal and C-terminal domains of the N protein of SARS-Cov2, and is capable of displacing bound nucleic acids, and of disrupting LLPS. It is concluded that HCQ forms a viable drug target.

The authors use a DNA ligand in this study, which they describe as stem-loop but also as single stranded. Presumably this means that at least part is double stranded? Please explain. The biological ligand is RNA. Please provide justification for why DNA is an appropriate molecule to choose, and why RNA was not used.

The NMR data are fairly convincing that there is specific binding to both domains. The binding is however very weak. The authors’ use of molar ratio in their presentations is perfectly reasonable, but hides the fact that it takes a very high concentration of HCQ to cause a useful effect: if they are using 100 uM or 200 uM domain, and only seeing reductions in binding with ratios of around 10-fold, then they need HCQ concentrations of at least 1 mM to see any useful reduction in binding. This point needs making clearly in the manuscript.

In some of the chemical structures (Fig S1, Fig S7) some of the letters and numbers are reversed. Please correct these.

Figs S2, S3, S4, S5 present NMR spectra showing loss of signal intensity due to S2m binding, which then recovers on addition of HCQ. A good indication of binding is that the effects on the spectrum should be specific, ie some peaks shift or reduce in intensity while others do not. My impression from these spectra is that all the peaks simply reduce in intensity fairly uniformly. It is clearly true that some peaks remain red for longer, but these are peaks that typically are sharp and intense anyway. The authors should present an analysis of signal loss across the peaks in the spectrum, to show if there is any clear evidence for specific loss of intensity rather than a general loss of intensity everywhere. In the text, the authors comment (page 7) that the loss of intensity is due to an intermediate exchange binding. If my interpretation is correct, then actually it is just protein precipitation or aggregation (eg in LLPS), which need not reflect any direct binding at all, and certainly says nothing about the timescale. So the authors need to analyse and present their data more carefully.

---

## [Reviewer Report]

*Comments to Author*: Reviewer #1: This paper describes experiments, mostly using NMR titrations, that are used to conclude that hydroxychloroquine (HCQ) binds to both the N-terminal and C-terminal domains of the N protein of SARS-Cov2, and is capable of displacing bound nucleic acids, and of disrupting LLPS. It is concluded that HCQ forms a viable drug target.

The authors use a DNA ligand in this study, which they describe as stem-loop but also as single stranded. Presumably this means that at least part is double stranded? Please explain. The biological ligand is RNA. Please provide justification for why DNA is an appropriate molecule to choose, and why RNA was not used.

The NMR data are fairly convincing that there is specific binding to both domains. The binding is however very weak. The authors’ use of molar ratio in their presentations is perfectly reasonable, but hides the fact that it takes a very high concentration of HCQ to cause a useful effect: if they are using 100 uM or 200 uM domain, and only seeing reductions in binding with ratios of around 10-fold, then they need HCQ concentrations of at least 1 mM to see any useful reduction in binding. This point needs making clearly in the manuscript.

In some of the chemical structures (Fig S1, Fig S7) some of the letters and numbers are reversed. Please correct these.

Figs S2, S3, S4, S5 present NMR spectra showing loss of signal intensity due to S2m binding, which then recovers on addition of HCQ. A good indication of binding is that the effects on the spectrum should be specific, ie some peaks shift or reduce in intensity while others do not. My impression from these spectra is that all the peaks simply reduce in intensity fairly uniformly. It is clearly true that some peaks remain red for longer, but these are peaks that typically are sharp and intense anyway. The authors should present an analysis of signal loss across the peaks in the spectrum, to show if there is any clear evidence for specific loss of intensity rather than a general loss of intensity everywhere. In the text, the authors comment (page 7) that the loss of intensity is due to an intermediate exchange binding. If my interpretation is correct, then actually it is just protein precipitation or aggregation (eg in LLPS), which need not reflect any direct binding at all, and certainly says nothing about the timescale. So the authors need to analyse and present their data more carefully.

Reviewer #2: This is a very well conducted study with interesting results. The structures are well presented and the NMR work carefully done.

My minor revision suggestion is about some minor language problems- the ms should be scrutinized again before publication.

examples of some problems:

p.6, 2nd para., line 4:... NTD characteristic of broadening of HSQC... (difficult to understand)

p. 10, 1st para., line 8:...valueless... (do you mean valuable?)

p. 11, 3rd para., line 2:.. effective... (do you really mean effective?)

In figures 2 and 4 you call one color "brown", but in my print it more looks like yellow (?)

---

## [Reviewer Report]

*Comments to Author*: This paper reports a novel and interesting observation, and so certainly fits into the remit of this journal. Because QRD takes papers that are slightly speculative, I have applied a slightly less rigorous standard of proof here than I would normally do. The authors have clearly demonstrated interactions between the NTD, S2m, and HCQ. How specific these are is much more difficult to determine. LLPS is by its nature a bulk phase separation rather than a specific binding event, and I therefore feel it would be unfair to look for too much evidence of specific intermolecular interactions. Overall I feel roughly 50% confident that there is a genuine and biologically relevant interaction going on here. Given the topic and the journal, I feel this is acceptable.

A key point is that the interaction is very weak: it requires concentrations of HCQ approaching 1 mM in order to see convincing effects. At the end of the paper, the authors note that joining together two HCQ molecules could produce stronger binding. This is true, and a helpful point, but nevertheless the authors should make it clear that the interactions that they are seeing are extremely weak, implying that the observations they make here are unlikely to provide an explanation for any possible therapeutic effect of HCQ. At the very least, they need to say this explicitly in the abstract and the discussion.

They quote their dissociation constants to one decimal place. These are far too precise given the errors in the determination: I suggest that for example 50 - 20 would be a more suitable precision.

Fig S2 shows the results of adding S2m to NTD. Signals broaden, but I suspect this is due to incorporation of the NTD into an amorphous phase-separated aggregate. This implies that the peak broadening is not due to intermediate exchange (page 6) but to formation of a slowly tumbling aggregate. The text here should be changed. I suspect that all signals are decreasing in intensity together - is this true? If they are, then this is almost certainly formation of an aggregate.

On page 10, the authors describe small molecule drugs as "valueless". I suspect they mean exactly the opposite (ie valuable, or the confusing but acceptable word invaluable).

Page 12 line 2:"Kd value is the time" - the word time should be product.

In Figure 2, can the authors explain the significance of the colours used in (A).

Supplementary figures S1 and S7 have structures drawn upside down. Please change them.

---

## [Reviewer Report]

*Comments to Author*: Reviewer #1: This paper reports a novel and interesting observation, and so certainly fits into the remit of this journal. Because QRD takes papers that are slightly speculative, I have applied a slightly less rigorous standard of proof here than I would normally do. The authors have clearly demonstrated interactions between the NTD, S2m, and HCQ. How specific these are is much more difficult to determine. LLPS is by its nature a bulk phase separation rather than a specific binding event, and I therefore feel it would be unfair to look for too much evidence of specific intermolecular interactions. Overall I feel roughly 50% confident that there is a genuine and biologically relevant interaction going on here. Given the topic and the journal, I feel this is acceptable.

A key point is that the interaction is very weak: it requires concentrations of HCQ approaching 1 mM in order to see convincing effects. At the end of the paper, the authors note that joining together two HCQ molecules could produce stronger binding. This is true, and a helpful point, but nevertheless the authors should make it clear that the interactions that they are seeing are extremely weak, implying that the observations they make here are unlikely to provide an explanation for any possible therapeutic effect of HCQ. At the very least, they need to say this explicitly in the abstract and the discussion.

They quote their dissociation constants to one decimal place. These are far too precise given the errors in the determination: I suggest that for example 50 - 20 would be a more suitable precision.

Fig S2 shows the results of adding S2m to NTD. Signals broaden, but I suspect this is due to incorporation of the NTD into an amorphous phase-separated aggregate. This implies that the peak broadening is not due to intermediate exchange (page 6) but to formation of a slowly tumbling aggregate. The text here should be changed. I suspect that all signals are decreasing in intensity together - is this true? If they are, then this is almost certainly formation of an aggregate.

On page 10, the authors describe small molecule drugs as "valueless". I suspect they mean exactly the opposite (ie valuable, or the confusing but acceptable word invaluable).

Page 12 line 2:"Kd value is the time" - the word time should be product.

In Figure 2, can the authors explain the significance of the colours used in (A).

Supplementary figures S1 and S7 have structures drawn upside down. Please change them.